# Bispecific T-Cell Engagers Therapies in Solid Tumors: Focusing on Prostate Cancer

**DOI:** 10.3390/cancers15051412

**Published:** 2023-02-23

**Authors:** Diana C. Simão, Kevin K. Zarrabi, José L. Mendes, Ricardo Luz, Jorge A. Garcia, William K. Kelly, Pedro C. Barata

**Affiliations:** 1Department of Medical Oncology, Centro Hospitalar Universitário de Lisboa Central, 1169-050 Lisbon, Portugal; 2Department of Medical Oncology, Sidney Kimmel Cancer Center, Thomas Jefferson University, Philadelphia, PA 19107, USA; 3Division of Solid Tumor Oncology, University Hospitals Seidman Cancer Center, Case Western Reserve University, Cleveland, OH 44106, USA

**Keywords:** prostate cancer, immunotherapy, bispecific T-cell engagers, prostate-specific membrane antigen, prostate stem cell antigen

## Abstract

**Simple Summary:**

Cancer treatments have significantly changed with the introduction of immunotherapy. Recently, the development of new agents that harness the redirection of T-cells against cancer is rapidly emerging in multiple tumor types. Since bispecific T-cell engager (BiTE) therapies have demonstrated clinical benefit in hematologic malignancies, their application to solid tumors has been an active area of investigation. However, in prostate cancer, due to the heterogeneous and immune-suppressive tumor microenvironment, the development of immunotherapy strategies remains a therapeutic challenge. In this review, we summarize the current development of BiTE therapies in solid tumors with a particular focus on clinical trials in advanced prostate cancer.

**Abstract:**

Over the past decade, immunotherapy has demonstrated an impressive improvement in treatment outcomes for multiple cancers. Following the landmark approvals for use of immune checkpoint inhibitors, new challenges emerged in various clinical settings. Not all tumor types harbor immunogenic characteristics capable of triggering responses. Similarly, many tumors’ immune microenvironment allows them to become evasive, leading to resistance and, thus, limiting the durability of responses. To overcome this limitation, new T-cell redirecting strategies such as bispecific T-cell engager (BiTE) have become attractive and promising immunotherapies. Our review provides a comprehensive perspective of the current evidence of BiTE therapies in solid tumors. Considering that immunotherapy has shown modest results in advanced prostate cancer to date, we review the biologic rationale and promising results of BiTE therapy in this clinical setting and discuss potential tumor-associated antigens that may be integrated into BiTE construct designs. Our review also aims to evaluate the advances of BiTE therapies in prostate cancer, illustrate the major obstacles and underlying limitations, and discuss directions for future research.

## 1. Introduction

Cancer immunotherapy, defined as the science of modulating the immune system against cancer, represents a new paradigm shift in the field of oncology, prolonging survival in several solid tumors [1]. Nevertheless, it took decades of basic science discoveries to demonstrate the ability of modulating the immune system to treat cancer and subsequently establish its role in clinical practice [2]. 

Since immune response against cancer involves complex interactions between tumor, host, and environment, different strategies have been developed including immunostimulatory cytokines, vaccines, adoptive cell therapies, oncolytic viruses, and immune checkpoint inhibitors (ICIs) [3]. High-dose interleukin-2 (IL-2), a potent inducer of cytotoxic T-cells and NK cells, was one of the first FDA-approved immunotherapy drugs in advanced cancer, with a role in the treatment of melanoma and renal cell carcinoma [4]. Although high-dose cytokine therapy is no longer used due to its short half-life and significant toxicity in therapeutic doses, the activity of IL-2 provided a fundamental understanding of the therapeutic potential of T-cell regulation in the development of new immunotherapy strategies [5]. 

ICIs targeting CTLA-4 and PD1/PD-L1 pathways emerged as a revolutionary cancer treatment strategy, due to impressive clinical responses and overall outcome benefits in certain tumor types [6,7]. However, the efficacy of ICIs in prostate cancer have been modest, except for mismatch-repair-deficient or microsatellite-instability-high tumors, in which pembrolizumab has been approved in a tumor-agnostic manner [8]. Moreover, the development of therapeutic cancer vaccines led to the approval of sipuleucel-T in metastatic prostate cancer. Despite its proven benefit in overall survival [9], questions remain about its true clinical benefit and its role within the treatment paradigm of metastatic prostate cancer. 

Recently, novel immunotherapies that redirect T-cells against tumor antigens through antibody fragments independent of major histocompatibility complex (MHC) presentation have been under investigation. In particular, chimeric antigen-receptor-modified (CAR) T-cells and bispecific T-cell engagers (BiTEs) have demonstrated remarkable clinical responses in hematologic malignancies [10,11].

Nonetheless, the development of these novel T-cell redirection approaches in solid tumors still presents major obstacles that limits its clinical application, including tumor heterogeneity and off-tumor toxicity [12,13].

This comprehensive review aims to describe the current evidence of BiTE therapies in solid tumors with a focus on ongoing clinical trials in the treatment of advanced prostate cancer.

## 2. Bispecific T-Cell Engagers (BiTEs) in Cancer Treatment

Bispecific antibodies (bsAb) were first described by Nisonoff and colleagues in the 1960s, as an antibody-based molecule with two distinct antigen-binding sites, which can function to physically bridge two different cells [14]. By simultaneously binding an antigen on tumor cells and a surface molecule on T-cells, bsAbs can redirect and activate T-cells to induce tumor lysis [15,16].

Since the 1980s, multiple bsAb formats have been developed [15]. In preclinical models, these early constructs showed relatively limited efficacy, with high drug concentrations, high effector-to-target ratios required to induce T-cell-mediated cancer cell lysis, and showed significant “off-target” toxicity [17]. However, novel strategies are evolving to overcome these limitations in order to expand and further optimize bsAb formats.

BsAbs are categorized based on their structure and mechanisms of action, specificity, and affinity for target antigen [18]. From a structural point of view, bsAbs are classified by the presence/absence of the Fc region [19]. BiTEs represent prototypical Fc-free bsAbs, with several new constructs currently under clinical evaluation in solid tumors. 

### 2.1. BiTE Design and Mechanism of Action

BiTEs are recombinant proteins composed of two different single-chain variable fragment (scFv) regions from two different monoclonal antibodies. The scFv constructs are covalently connected by a flexible small peptide linker, altogether comprising a ~55 kDa polypeptide chain [17,19]. One scFv-binding domain is engineered to target a select tumor-associated antigen (TAA) and the other scFv domain is typically specific for CD3, the invariant component of the T-cell receptor (TCR) complex [20,21,22]. The length of the inter-scFv linker varies depending on construct, but linker size does not directly impact tumor-killing activity [23]. However, linker length must account for flexibility, stability, and the orientation of binding interaction between TAA and epitope [19,24].

In general, the TCR complex on the surface of T-cells recognizes antigens that are presented via the major histocompatibility complex (MHC) and this interaction triggers a signaling cascade, involving transcription factor activation and cytoskeletal remodeling, resulting in T-cell activation [20]. BiTEs are unique in their ability to redirect T-cells against TAAs on tumor cells and directly activate T-cells, independent of TCR/MHC interaction [21]. More specifically, T-cell activation by interaction with the TCR complex engages T-cells to form an immune synapse on the surface of tumor cells, resulting in release of cytokines, perforins, and granzymes that induces cancer cell apoptosis [21,25]. Moreover, the activation of effector T-cells occurs only when both scFv-binding domains are engaged with their respective targets [26].

In comparison with alternative bsAbs formats and monoclonal IgG antibodies, BiTEs have a 100-to-10,000-fold higher efficacy in tumor cell lysis with a low ratio of T-cells to target tumor cells in cellular models [27]. Subsequent to BiTE-induced T-cell activation, the diffusion of released cytokines in the immune synapse also plays a role in upregulation of cell surface molecules of the surrounding cells, further contributing to the antitumor activity of BiTEs, commonly named the “bystander effect” [28]. 

Furthermore, BiTEs can be produced in large quantities by mammalian cell lines, minimizing interpatient variability and providing “off-the-shelf” therapies that now are undergoing investigation in a multitude of tumor types [29].

### 2.2. Blinatumomab, the First BiTE Construct in Clinical Practice

The CD19/CD3 BiTE molecule blinatumomab has served as clear proof of concept of antitumor activity and clinical efficacy of T-cell engagers in B-cell malignancies. Blinatumomab’s clinical efficacy and favorable safety profile lead to its first-in-class approval by the European Medicines Agency (EMA) and the U.S. Food and Drug Administration (FDA) for the treatment of both children and adults with relapsed or refractory Philadelphia chromosome (Ph)-negative precursor B-cell acute lymphoblastic leukemia (B-ALL) [30,31]. In the TOWER trial, a multicenter, international, phase 3 clinical trial, blinatumomab demonstrated a significant improvement in overall survival (7.7 months vs. 4.0 months; HR 0.71; 95% CI 0.55–0.93) and higher rates of hematological remission, compared to standard-of-care chemotherapy [32]. The FDA also approved blinatumomab for relapsed or refractory Ph-positive B-ALL, based on the ALCANTARA trial results that demonstrated remarkable long-term durability of responses in this setting [33]. Long-term follow-up data show that Blinatumomab treatment can render a complete response with minimal residual disease (MRD) in approximately three-quarters of treated patients [33,34], resulting in approval for treatment of MRD-positive patients with B-ALL.

Despite its efficacy, blinatumomab has also been associated with significant adverse events including cytokine release syndrome (CRS) and neurotoxicity, which may be life-threatening [35]. Treatment of patients requires inpatient hospitalization at time of drug administration for monitoring and to allow for prompt management of these potential events.

### 2.3. BiTE Therapy Safety Considerations: Cytokine Release Syndrome and Neurotoxicity

As the first marketed therapy within its class, blinatumomab’s safety profile and dose-limiting toxicities are well-described [35]. The two main toxicity concerns associated with BiTE immunotherapy correlate with its mechanism of action and include cytokine release syndrome (CRS) and neurotoxicity. 

CRS is an uncontrolled systemic inflammatory response characterized by high levels of pro-inflammatory cytokines, most notably interferon (IFN)-gamma, IL-1, and IL-6, and is induced by T-cell activation [36]. CRS clinical manifestations and severity varies from mild fever or rash to severe multi-organ failure [36]. When CRS occurs after dose administration of blinatumomab, symptoms usually appear during the first infusion cycle but can be delayed by days, and the risk of grade ≥3 CRS ranges from 2 to 11% for B-cell malignancies [32,33,34,35]. A higher incidence of CRS has been associated with higher tumor burden and drug dosage [36,37]. Early intervention is critical to prevent progression to life-threatening toxicity. In patients with mild CRS, supportive care is indicated, while grade ≥3 CRS is managed with infusion interruption and immunosuppression with glucocorticoids [38]. Tocilizumab, a humanized monoclonal antibody that targets the IL-6 receptor, was approved for the management of severe or life-threatening CRS [38]. To reduce the incidence of CRS, prophylactic use of dexamethasone combined with step-dosing administration of blinatumomab is recommended [36,37].

Similar to CRS, immune effector-cell-associated neurotoxicity syndrome (ICANS) pathophysiology is complex and incompletely understood and seems to be related to the production of pro-inflammatory cytokines, with subsequent T-cells adhesion to brain endothelium and disruption of the blood–brain barrier [39,40]. The incidence of grade ≥3 neurotoxicity with blinatumomab ranges from 5.5 to 24% [31,32,33,34]. Neurotoxicity occurs most commonly in the first treatment cycle and risk increases with higher doses of blinatumomab. The most common symptoms include dizziness, tremor, confusional state, and encephalopathy [41]. Management of ICANS require treatment interruption and corticosteroids; however, definitive evidence is lacking as to whether corticosteroids have a beneficial effect on the severity or duration of ICANS [39].

In addition, other relevant adverse events related to the CD19-targeting mechanism have been reported with blinatumomab, namely long-term B-cell aplasia and hypogammaglobulinemia [33,34,35]. As such, immunoglobulin replacement and prophylactic antibiotics should be considered on an individual case-by-case basis.

### 2.4. Limitations of BiTE Therapies and Innovative Strategies to Enhance Efficacy

A practical limitation of the prototypical BiTE molecule in clinical practice is their short half-life and the need for continuous intravenous administration [17]. New approaches to optimize drug delivery and alter pharmacokinetics include half-life-extended (HLE) BiTEs [42,43], which are single-chain polypeptides incorporating an additional Fc region, creating a bsAb with a higher molecular weight and extended half-life. With regards to alternate routes of administration, subcutaneous BiTEs have been investigated, showing a manageable safety profile similar to that previously reported for intravenous formulations [44].

“Off-the-shelf” BiTE manufacturing is a major advantage of this treatment modality, supporting its clinical applicability and cost-effectiveness, since large quantities are produced, without interpatient variability [29]. However, identification of target antigens that are ubiquitously expressed on tumor cells in all patients has been a critical challenge for the application of BiTE therapy to certain cancers, particularly solid malignancies. Many tumor-specific antigens are intracellular and are not accessible for standard T-cell engagers, while numerous cell-surface TAAs overexpressed in solid tumors lack high specificity and are often found at low levels in normal tissue [21,22]. As a consequence, “on-target, off-tumor” toxicity has been a challenge in TAA selection for solid tumors [45]. The balance between maximizing therapeutic potential of BiTEs while mitigating toxicity remains an area which requires further investigation [42].

Acquired treatment resistance to BiTE therapy is yet another limitation. Downregulation or loss of TAA expression has been described as a major mechanism of resistance to BiTE therapies [46]. This observation gave rise to the development of multiple novel T-cell engager constructs, with different pharmacokinetic and pharmacodynamic profiles, including different formats with higher stability in a construct that enables optimal interaction between the target and effector cell (dual affinity retargeting (DART^®^) bispecific antibodies), and some simultaneously targeting different TAAs (simultaneous multiple interaction T-cell engagers (SMITEs)) [17].

Another mechanism of resistance to BiTE therapy is the upregulation of inhibitory immune checkpoints within the tumor microenvironment (TME) [47]. The TME of solid tumors contains a complex composition of malignant, immune, and stromal cell populations that can suppress antitumor T-cell responses, which negatively affects T-cell engager efficacy [48]. To overcome this limitation, constructs with concomitant immune-checkpoint action (checkpoint inhibitor T cell-engagers) that target the PD-1/PD-L1 axis are under development, as well as combination strategies combining BiTEs with ICIs [49].

## 3. BiTEs in Solid Tumors

Despite the impressive results of BiTEs in hematological malignancies, illustrated by blinatumomab efficacy in B-ALL, first-generation compounds have failed to demonstrate significant antitumor activity in solid tumors [13,50]. Catumaxomab was the first bispecific T-cell engager approved by the EMA in 2009 to treat malignant ascites of epithelial cancers [51]. It is a trifunctional bispecific IgG antibody, with one arm recognizing the epithelial cell adhesion molecule (EpCAM) on tumoral cells and another arm targeting the CD3 subunit on T-cells. Furthermore, the functional Fc fragment binds to different immune accessory cells, such as monocytes, macrophages, and natural killer (NK) cells, inducing T-cell activation and NK cell recruitment [52]. EMA approval was based on an improvement of puncture-free survival and signs and symptoms of ascites from a large, randomized phase 2/3 trial [51]. However, intravenous administration of catumaxomab was associated with severe adverse events, including CRS and dose-dependent liver toxicity [53], with one patient experiencing fulminant fatal acute liver failure that led to the early termination of the trial, and later withdrawal of catumaxomab from the market.

Solitomab is another first-generation BiTE targeting EpCAMxCD3 (MT110 or AMG110) that was investigated in a phase 1 trial of 65 patients with relapsed/refractory advanced-stage solid cancers. Treatment was associated with dose-limiting toxicities, including severe diarrhea and increased liver enzymes, which precluded dose escalation to potentially therapeutic levels [54].

Subsequent next-generation BiTE molecules have been constructed which are directed against TAAs with reduced expression in non-neoplastic tissue and employ formats that do not include an Fc domain.

Specifically, in CEA-positive solid tumors such as metastatic colorectal cancer, RO6958688 (also known as CEA CD3 T-cell bispecific or RG7802) was administered as monotherapy or in combination with atezolizumab in a phase 1 trial that enrolled 35 patients. Antitumor activity was observed in monotherapy, with two patients showing partial response, which was enhanced when in combination with atezolizumab, and with a manageable safety profile [55].

More recently, preliminary data from the DUET-1 phase 1 trial showed that tidutamab (XmAb18087), a BiTE-targeting somatostatin receptor 2 (SSTR2), was well tolerated with a best overall response of stable disease in patients with advanced neuroendocrine tumors [56].

We are now witnessing increasing numbers of bispecific-based T-cell engagers undergoing rapid development and evaluation in several tumor types (Table 1). Prostate cancer target-antigens and clinical trials will be further discussed separately. 

Recently, the FDA and EMA approved a bispecific fusion protein indicated for the treatment of adult patients with HLA-A*02:01-positive metastatic uveal melanoma [70,71]. Tebentafusp is a first-in-class immune-mobilizing monoclonal T cell receptor (TCR) against cancer (ImmTAC^®^), comprising a TCR domain that binds with high affinity to a gp100 peptide presented by human leukocyte antigen—A*02:01 (HLA-A*02:01) on the cell surface of uveal melanoma tumor cells, and an effector domain which binds to the CD3 receptor on polyclonal T-cells. Tebentafusp significantly improved overall survival in patients with previously untreated metastatic uveal melanoma in a large, randomized, phase 3 study that led to its regulatory approval [72].

## 4. BiTEs in Advanced Prostate Cancer

The success of immunotherapy in treatment of advanced prostate cancer has been modest, as most modern immunotherapies have failed to achieve long-term remissions. To date, sipuleucel-T is the only approved immunotherapy for metastatic prostate cancer, yet it is not considered a cornerstone therapy for men with metastatic castration-resistant prostate cancer (mCRPC). Sipuleucel-T incorporates autologous antigen-presenting cells (APCs), with a recombinant fusion protein (PA2024), consisting of a prostate antigen, prostatic acid phosphatase (PAP), and granulocyte–macrophage colony-stimulating factor (GMCSF). Although survival benefit was demonstrated in patients with mCRPC [9], questions about the true mechanism of action of this agent still remain [73], thus limiting its application in daily clinical practice.

Except for tumors associated with microsatellite instability, most mCRPC tumors are considered immunologically “cold”, due to lack of pro-inflammatory cytokine production, sparse T-cell infiltration, and predominance of suppressive immune components [74,75]. To characterize tumor microenvironment of bone metastasis of prostate cancer, Kfoury et al. performed single-cell analysis and found bone marrow infiltration of tumor-associated macrophages and monocytes with overexpression of cytokine CCL2, leading to T-cell exhaustion as a mechanism of immunosuppression [76].

Furthermore, recent findings described T-cell-intrinsic androgen activity as a novel mechanism of resistance to immunotherapy [77].

Novel immunotherapies using MHC-independent T-cell redirection and activation have been an active area of research with hopes to overcome the immunosuppressive TME within prostate cancer [78]. Particularly in mCRPC, tumor-specific surface markers with relatively low expression in normal tissues have been investigated as potential TAA targets of novel T-cell redirection strategies. Currently, there are several targets for BiTE therapies under development in the prostate cancer disease space (Table 2), including constructs targeting prostate-specific membrane antigen (PSMA), prostate stem cell antigen (PSCA), six-transmembrane epithelial antigen of the prostate (STEAP-1), and Notch ligand delta-like protein 3 (DLL3).

### 4.1. BiTEs Targeting PSMA

PSMA is a type II transmembrane protein that is highly expressed on the surface of malignant prostate tissue, with variable low expression in non-neoplastic prostate tissue. Non-malignant tissues expressing PSMA include kidney proximal tubules, salivary glandular cells, and the gastrointestinal tract [94,95]. Additionally, PSMA is also expressed on the tumor-associated neovasculature of different solid malignancies [96,97], increasing the research focus on PSMA as TAA in non-prostate cancer therapies.

In addition, PSMA has been widely explored as a biomarker of prostate cancer activity for disease imaging, using radioactive PSMA tracers [98,99], and as a disease-specific target in the field of PSMA theranostics [100,101].

#### 4.1.1. Pasotuxizumab (BAY2010112 or AMG 212)

The first prototypical PSMA-targeting BiTE was pasotuxizumab (BAY2010112 or AMG 212). In a phase 1 clinical trial (NCT01723475), safety and maximum tolerated dose (MTD) of pasotuxizumab in mCRPC was evaluated. As part of an interim data monitoring analysis, it was reported that all 31 patients receiving subcutaneous (SC) injection had developed antidrug antibodies (ADAs) and prophylactic dexamethasone had no effect in mitigating this event [79]. The SC route of administration was discontinued, and the study proceeded with the continuous intravenous (cIV) infusion cohort only [80]. Hummel et al. reported a >50% PSA reduction in nine and three patients in the SC and cIV cohorts, respectively, including two long-term responders [79,80]. No grade 5 adverse events (AE) were reported. Nevertheless, all patients had at least one AE, including fever (85%), chills (38%), and fatigue (34%) [79]. MTD was not reached due to premature termination of the trial in favor clinical study of a next-generation BiTE construct targeting PSMA, AMG-160.

#### 4.1.2. Acapatamab (AMG-160)

Preclinical studies with a half-life extended BiTE, acapatamab (formerly AMG-160), demonstrated T-cell activation in human samples and upregulation of PD-L1 [102,103]. Given the interest in the development of next-generation PSMA-targeting BiTEs, acapatamab is further being assessed in early-phase clinical study. In an ongoing phase 1 clinical trial (NCT03792841) enrolling 43 patients with mCRPC, dose exploration of acapatamab is being evaluated in monotherapy and in combination with pembrolizumab [81,82]. Preliminary results of monotherapy, in a heavily pre-treated population (with a median of four prior lines of therapy), showed PSA reductions >50% occurred in 12/35 (34.3%) evaluable patients at data cut-off (July 2020) [81]. The majority of patients (n = 41; 95.3%) experienced an any-grade AE, including grade 3 CRS in 11 patients (25.6%). No grade 5 events or treatment discontinuation has been observed, although MTD has not yet been reached [81]. An exploratory phase of this trial assessing the combination of acapatamab with pembrolizumab is in progress.

Alternative combination therapy strategies with acapatamab are also being evaluated. An elegantly designed phase 1/2 trial (NCT04631601) consisting of three subprotocols evaluating safety, tolerability, and MTD of acapatamab, in combination with enzalutamide, abiraterone, or AMG 404, a monoclonal antibody targeting PD-1 receptor, is currently recruiting patients [83].

#### 4.1.3. Novel Emerging Constructs Targeting PSMA

ES414/APVO4141/MOR209 is a bispecific antibody constructed with an ADAPTIR^®^ format, which contains a modified antibody Fc region that improves serum stability but does not cross-link T-cells or target cells and two scFv fragments each targeting PSMA and CD3 [84]. In comparison with prototypical BiTE formats, it has a prolonged half-life; however, high immunogenicity of this construct gave rise to unacceptable systemic toxicity and to early discontinuation of the clinical trial. 

In contrast with previous BiTE structural design, HPN424 is a trispecific antibody (TriTAC^®^) designed with three binding domains. In addition to PSMA and CD3 targeting, it incorporates an albumin-binding domain to prolong serum half-life and increase stability [104]. Data from a phase 1/2a clinical trial (NCT03577028) including men with mCRPC who have received more than two prior systemic therapies, showed that 21% of patients had post-baseline PSA declines and reduction in circulating tumor cells (CTC) occurred in 32 of 56 pts (57%) with measurable CTC at baseline. Most common grade >3 AE were AST increase (18%), anemia (11%), and ALT increase (11%). Of note, grade 3 CRS was observed in 4% of patients, occurring with first dose administration, and MTD was not reached [85]. 

JNJ-63898081/JN-081 is a novel bispecific antibody targeting PSMA and CD3 engineered by a next-generation technology platform denominated DuoBody^®^. This technology is believed to produce more stable bsAb constructs and retain endogenous IgG structure and pharmacokinetics possibly leading to improved tolerability and efficacy. In a phase 1 trial, all 39 patients enrolled experienced at least one treatment-emergent AE, most commonly pyrexia (n = 27; 69.2%) and CRS (n = 26; 66.7%). Grade 2 CRS was observed at higher doses and was partially mitigated by SC and step-up dosing, with no reported grade ≥3 CRS. Transient PSA decreases were observed at treatment doses greater than 30 µg/kg SC [86].

CC-1 is a bsAb targeting PSMA with a unique IgG-based structure. CC-1 was developed in an IgGsc format (IgG molecule with two c-terminal single chain moieties), which includes an Fc domain of an IgG1 antibody linked to two scFv-binding domains targeting PSMA and CD3. The modified Fc domain is expected to prolong its serum half-life, as well as lower immunogenicity and thereby mitigate toxicity. Despite preliminary data of 14 patients from a dose-escalation phase 1 trial which showed that the majority of patients suffered from a CRS event (79% of patients), the CRS did not exceed grade 2 and resolved in most cases without the need for administration of tocilizumab. Results from a dose expansion of this phase 1 clinical trial are highly anticipated and the study is currently enrolling patients (NCT04104607) [87].

Additional ongoing trials concerning bispecific antibodies targeting PSMA are summarized in Table 2.

### 4.2. Other Potential TAAs in Prostate Cancer

#### 4.2.1. Prostate Stem Cell Antigen (PSCA)

PSCA is a cell-surface glycoprotein encoded by the PSCA gene, which is overexpressed in prostate gland cells as well as urothelial, pancreatic, renal, and non-small cell lung cancer [105,106]. Although its biological function is not completely understood, PSCA has been associated with advanced disease and poor prognosis in prostate cancer [107].

An open-label, dose escalation clinical trial of GEM3PSCA (NCT03927573), a PSCAxCD3 bispecific antibody (ATAC^®^ format) is enrolling patients with PSCA-expressing tumor types after failure of standard therapy.

#### 4.2.2. Six-Transmembrane Epithelial Antigen of the Prostate-1 (STEAP-1)

Acting as a membrane channel or transporter protein in cell junctions of epithelial cells, STEAP-1 is overexpressed on the surface of prostate cancer cells with low or no expression on normal tissue [108]. 

Binding simultaneously to STEAP-1 and CD3, AMG 509 is a bispecific T-cell engager (XmAb^®^ format) that is being evaluated in a phase 1 trial (NCT04221542) as monotherapy and in combination with enzalutamide or abiraterone [91].

### 4.3. Neuroendocrine Prostate Cancer

De novo neuroendocrine prostate cancer (NEPC) is a rare tumor, corresponding to less than 2% of all cancers at the time of diagnosis [109]; however, treatment-related-NEPC is found in 10.5–17% of patients with mCRPC after treatment with androgen signaling inhibitors [110].

The 2022 WHO Classification of Tumors of the Urinary and Male Genital System describes treatment-related neuroendocrine prostatic carcinoma (t-NEPC) as a distinct entity: “tumors demonstrating complete neuroendocrine differentiation or partial neuroendocrine differentiation with adenocarcinoma following androgen deprivation therapy” [111]. Regarding histological and immunological features, some are pure small cell, or less commonly large cell, neuroendocrine carcinoma, while others are mixed tumors with a component of high-grade adenocarcinoma [112].

Although it is unclear whether de novo NEPC and t-NEPC have a shared clonal origin, emerging evidence suggests that adenocarcinoma to NEPC transdifferentiation may be driven by concomitant inactivation of TP53 and RB1, and alterations of epigenetic regulation and transcription factors [113].

Initially identified as a surface protein overexpressed in small cell lung cancer, DLL3 is also highly expressed in neuroendocrine prostate cancer (NEPC) [114], emerging as a potential candidate to target for T-cell redirecting therapies since no standard treatment approach for NEPC exists and it remains an unmet need.

Tarlatamab or AMG 757 is an HLE BiTE (DLL3xCD3) that is being evaluated in a phase 1b trial (NCT04702737), which is recruiting patients with metastatic de novo or treatment-emergent NEPC [87].

Novel TAAs for NEPC have been investigated. Han et al. suggested that KIT pathway inhibition may be a potential target in NEPC treatment [115].

### 4.4. Overcoming Hurdles to Successful Implementation of BiTE Therapy within the mCRPC Treatment Paradigm

Successful development of BiTEs requires a tumor-restricted TAA which allows for effective antigen selectivity and minimal “off-tumor, on-target” toxicity. Prostate cancer is an ideal disease setting for BiTE development given the myriad of unique TAAs that have been discovered. Initial issues with drug immunogenicity and rapid drug clearance have been addressed with the advent of HLE BiTEs. Although toxicity in the early-generation prostate cancer BiTE trials has been ubiquitous, it has also been manageable. Our understanding of BiTE toxicity is informed from prior experiences with hematologic malignancies, and early intervention at first sign of CRS with systemic corticosteroids and immunosuppression has proven to be an effective approach at safely achieving greater therapeutic dose thresholds for optimal drug delivery.

As we optimize the BiTE format in respect to pharmacokinetics, tissue selectivity, and toxicity, we are also in need of TME modulation to allow BiTE effector function within a less immunologically ‘cold’ and suppressive TME. Multimodal therapies with BiTEs in combination with novel modalities are a strategy under investigation. We now have a clinical study employing BiTEs in combination with ICIs in prostate cancer, which is a strategy with strong biologic rationale as disinhibiting immune effector cells can help augment BiTE response. 

The trials to date in prostate cancer employ a heavily pre-treated patient population, as is the case of BiTE trials in other solid-tumor malignancies. In patients with mCRPC, prior treatment with anti-androgen therapy, theranostics therapy, and typically multiple lines of cytotoxic chemotherapy may not be an ideal sequence prior to T-cell immunotherapy. Development of castration resistance after anti-androgen therapy, as well as usage of cytotoxic chemotherapy, each accelerate development of an exhausted T-cell phenotype rich in T-regulatory cells, which is not conducive for immune response. As such, earlier integration of T-cell immunotherapy and early referral for trial consideration is recommended. 

## 5. Future Perspectives 

The underlying causes of the limited effectiveness of BiTE therapy in solid-tumor malignancies are multifactorial. It is unlikely that a single strategy to optimize BiTE construct design will dramatically improve treatment efficacy. Ongoing clinical investigation into the immune-escape mechanisms and better characterization of the immune milieu within the TME of each respective tumor type will be critical in order to identify novel target antigens which can then be integrated into next-generation BiTE construct design. Moreover, the success of BiTE therapy in solid-tumor malignancies will likely rely on two key additional factors: (1) mitigation of severe toxicity and (2) ideal identification and selection of patients who may benefit most from immunologic response.

BiTE therapy toxicity, namely CRS, is highly predictable and some form of systemic inflammatory response is expected in nearly all patients. Early identification of toxicity and intervention with corticosteroids has certainly proven efficacious in preventing severe toxicity leading to end-organ dysfunction. There is ongoing concern that immunosuppressive intervention for CRS will counteract any anti-tumor immune response triggered by therapy. However, recent studies evaluating early administration of tocilizumab with a newly developed PSMAxCD3 bsAb revealed reduction in undesired sequelae of CRS without affecting therapeutic activity [116]. Taking early measures to attenuate immune-related toxicity may prevent dose-reduction and early drug discontinuation, thereby allowing patients to benefit from more drug exposure and maximized treatment effect.

Ideal patient selection represents yet another challenge which may be an area of focus for future research. We now have identified a plethora of tumor-associated antigens (TAAs) that demonstrate high specificity for target antigen-expressing tumor, yet TAA as a biomarker has been insufficient in patient selection to identify exceptional responders. It is very possible that chemotherapy refractory and heavily pre-treated patient populations preclude optimal patient selection. The timing of BiTE therapy within the treatment cascade also requires further investigation and is an ongoing area of research [117]. Early introduction of BiTE therapy within the disease course may be a pathway to maximize therapeutic potential prior to development of an exhausted immune phenotype within the TME. 

Following the trend in recent years in the use of combination therapies, exploring the synergy between BiTEs and other immunomodulatory drugs may be an ideal strategy to improve efficacy in solid tumors. BiTE therapies in combination with immune checkpoint inhibitors are currently under investigation and may prove to be an effective strategy capable of augmenting immunogenic treatment effects. As we continue to refine our understanding of the tumor immune profile with spatial analyses utilizing genomics and transcriptomics, we will likely be able to strengthen the biologic rationale for combinations therapies and areas of synergy amongst classes of agents.

## 6. Conclusions

Immunotherapy has become an established cornerstone of therapy within the treatment paradigm of several tumor types. Novel T-cell redirecting strategies represent more contemporary immunotherapies, and BiTEs in particular have shown substantial efficacy in hematological malignancies. However, some factors challenge the implementation of BiTEs in solid tumors, namely the lack of target antigen expression, tumor inaccessibility, and the impact of an immunosuppressive tumor microenvironment. Prostate cancer in particular is known for its immune-desert phenotype, with little observed benefit from modern immunotherapies. However, several strategies are currently being investigated to improve BiTE application in an immunologically “cold” tumor and are poised to have transformative impacts within the prostate cancer disease space. Future results of ongoing studies in which the combination of BiTE with innovative therapies may provide some answers and paradigm-changing advances in upcoming years.

## Figures and Tables

**Table 1 cancers-15-01412-t001:** Bispecific T-cell engagers therapy in solid tumors (clinicaltrials.gov accessed on 1 November 2022).

NCT	Phase	Drug (Format)	Target	Indication	Status	Results	Ref.
EpCAM							
NCT00836654	2/3	Catumaxomab or Removab^®^ (Triomab^®^)	EpCAM xCD3	Malignant ascites and EpCAM-positive tumors	Completed	N = 258 (129 ovarian cancer) Puncture-free survival: 46 vs. 11 days AE: fever (60%); abdominal pain (43%)	[51]
NCT01065246	2	Catumaxomab	EpCAM xCD3	Malignant ascites due to epithelial carcinoma	Completed	N = 8 (rechallenge of intraperitoneal catumaxomab) Puncture-free survival: 47.5 days	[57]
NCT00326885	2	Catumaxomab	EpCAM xCD3	Malignant ascites ovarian cancer	Completed	N = 32 Puncture-free survival: 29.5 days Ascites symptoms improved	[58]
NCT01246440	2	Catumaxomab	EpCAM xCD3	Ovarian cancer	Completed	N = 46 (consolidation therapy) Median duration treatment: 13 days Grade 3–4 AE in 29 pts (74.4%) Treatment interruption in 4 (10.2%)	[59]
NCT00189345	2	Catumaxomab	EpCAM xCD3	Platinum refractory ovarian, fallopian tube, and peritoneal neoplasms	Completed	N = 46 (low dose 23 + high dose 22) No difference AE low vs. high Stable disease in 2 pts (low) and 5 pts (high)	[60]
NCT01815528	2	Catumaxomab	EpCAM xCD3	Recurrent epithelial ovarian cancer	Completed	Not reported	
NCT00563836	2	Catumaxomab	EpCAM xCD3	Ovarian cancer	Completed	Not reported	
NCT04222114	3	Catumaxomab	EpCAM xCD3	Gastric cancer	Recruiting		
NCT01504256	2	Catumaxomab + FLOT	EpCAM xCD3	Gastric adenocarcinoma with peritoneal carcinomatosis	Completed	N = 31 (FLOT + catumaxomab 15 pts (A) vs. FLOT alone 16 pts (B)) Complete remission of carcinomatosis: 27% (A) vs. 19% (B) (*p* = 0.69). Severe AE: fever (23%), abdominal pain (31%), elevated liver enzymes (31%). Median PFS: 6.7 (A) vs. 5.4 months (B) (*p* = 0.71).	[61]
NCT00464893	2	Catumaxomab	EpCAM xCD3	Gastric cancer	Completed	Not reported	
NCT00352833	2	Catumaxomab	EpCAM xCD3	Gastric cancer	Completed	Not reported	
NCT04501744	1	M701	EpCAM xCD3	Malignant ascites	Recruiting		
NCT00635596	1	Solitomab or MT110 or AMG110	EpCAM xCD3	Relapsed/refractory solid tumors	Completed	N = 65 (35 colorectal; 10 ovarian; 8 gastric; 6 NSCLC; 3 SCLC; 3 mCRPC) 95% Grade ≥ 3 AE, mainly diarrhea, elevated liver parameters and lipase	[54]
CEA							
NCT02324257 NCT02650713	1	RO6958688 or RG7802 + atezolizumab	CEA xCD3	CEA-positive tumors	Completed	N = 36 pts in monotherapy + 10 pts in combination Grade ≥ 3 AEs: infusion related reaction (16.3%) and diarrhea (5%)	[55]
NCT01284231	1	AMG211 or MEDI-565	CEA xCD3	Gastrointestinal adenocarcinomas	Completed	N = 39 (28 colorectal, 6 pancreatic, 5 other) Grade ≥ 3 AE in 5 pts (hypoxia n = 2, diarrhea, and CRS) Stable disease in 11 pts (28%)	[62]
NCT02291614	1	AMG211 or MEDI-565	CEA xCD3	Gastrointestinal adenocarcinomas	Completed	Terminated due to high immunogenicity at high doses of >3.2 mg	[63]
NCT03337698	1/2	RO6958688 + atezolizumab	CEA xCD3	NSCLC	Recruiting		[64]
EGFR							
NCT02620865	1/2	EGFR Bi-armed activated T-cells (BATs)	EGFR xCD3	Advanced pancreatic cancer	Completed	N = 7 No dose-limiting toxicities (DLTs), Median time to progression: 7 months	[65]
NCT03269526	1/2	EGFR BATs	EGFR xCD3	Advanced pancreatic cancer	Recruiting		
NCT03296696	1	AMG596	EGFRvIII xCD3	Glioblastoma	Completed	Not reported	[66]
NCT03344250	1	EGFR BATs + Temozolomide+ RT	EGFR xCD3	Glioblastoma	Active, not recruiting		
gpA33							
NCT02248805	1	MGD007 (DART^®^)	gpA33 xCD3	Metastatic CRC	Completed	Not reported	[67]
NCT03531632	1/2	MGD007 + MGA012	gpA33 xCD3	Metastatic CRC	Completed	Not reported	
HER2							
NCT04501770	1	M802	HER2 xCD3	HER2-positive advanced solid tumors	Not yet recruiting		
NCT03448042	1	Runimotamab + trastuzumab + tocilizumab	HER2 xCD3	Locally advanced or metastatic HER2-expressing solid tumors	Recruiting		
NCT03272334	1/2	HER2 BATs + Pembrolizumab	HER2 xCD3	Metastatic breast cancer	Recruiting		
Other							
NCT03411915	1	Tidutamab (XmAb18087)	SSTR2 xCD3	NET and GIST	Completed	N = 41 Grade ≥ 3 AE: lymphopenia (29.3%): transaminase and GGT increase (19.5%); hypophosphatemia (9.8%) and lipase increase (7.3%)	[56]
NCT04590781	1/2	Tidutamab (XmAb18087) + Pembrolizumab	SSTR2 xCD3	Advanced Merkel cell carcinoma and ES-SCLC	Completed	Not reported	
NCT04424641	1/2	GEN1044 (DuoBody^®^)	5T4 xCD3	Solid tumors	Completed	Results on submission clinicaltrials.gov (accessed on 1 November 2022)	
NCT05180474	1	GEN1047 (DuoBody^®^)	B7H4 xCD3	Solid tumors	Recruiting		
NCT04083599	1/2	GEN1042	4-1BB xCD40	Solid tumors	Recruiting		
NCT04496674	1	CC-1 + Tocilizumab	PSMA xCD3	NSCLC	Recruiting		
NCT04260191	1	AMG910	CLDN18.2 xCD3	Gastric and gastroesophageal junction adenocarcinoma	Active, not recruiting		
NCT03146637	2	Activated CIK	MUC1/CEA/EpCAM/GPC3xCD3	Advanced liver cancer	Recruiting		
NCT03319940	1	AMG757 (HLE) + Pembrolizumab	DLL3	SCLC	Recruiting		[68]
NCT04471727	1/2	HPN328 (TriTAC)	DLL3	SCLC	Recruiting		
NCT04590326	1/2	REGN4018 or REGN5668 + Cemiplimab	MUC16 xCD3 or MUC16 xCD28	Ovarian cancer, fallopian tube cancer, peritoneal cancer	Recruiting		[69]
NCT03564340	1/2	REGN4018 + Cemiplimab	MUC16 xCD3	Ovarian cancer, fallopian tube cancer, peritoneal cancer	Recruiting		
NCT04117958	1	AMG199 (HLE)	MUC17 xCD3	MUC17-positive solid tumors	Recruiting		

**Table 2 cancers-15-01412-t002:** Clinical trials of Bispecific T-cell engagers in prostate cancer. (clinicaltrials.gov accessed on 10 November 2022).

NCT	Phase	Drug (Format)	Target	Indication	Status	Results	Ref.
NCT01723475	1	Pasotuxizumab, BAY2010112 or AMG212	PSMA xCD3	mCRPC	Completed	N = 47 (31 sc + 16 iv) AE Grade 3: 53% MTD not reached due to early stop >50% PSA decline in 9 sc + 3 iv pts	[79,80]
NCT03792841	1	Acapatamab or AMG160 (HLE) + Pembrolizumab	PSMA xCD3	mCRPC	Active, not recruiting	N = 43 (monotherapy) Grade 3 CRS: 25.6% MTD not yet reached >50% PSA decline in 12/35 (34.3%)	[81,82]
NCT03792841	1	Acapatamab + Enzalutamide + Abiraterone + AMG 404	PSMA xCD3	mCRPC	Active, not recruiting		[83]
NCT02262910	1	ES414 or MOR209 or APVO411 (ADAPTIR^®^)	PSMA xCD3	mCRPC	Completed	Discontinued due to high immunogenicity of the construct	[84]
NCT03577028	1/2	HPN424 (TriTAC^®^)	PSMA xCD3	mCRPC	Unknown	N = 80 Grade 3 CRS: 4% MTD not yet reached PSA decline in 13/63 pts (21%), including 3 PSA50, 2 PSA30 responses.	[85]
NCT03926013	1	JNJ-63898081 or JNJ-081 (DuoBody^®^)	PSMA xCD3	mCRPC	Completed	N = 39 (27 sc + 12 iv) All pts ≥ 1 treatment-emergent AE No grade ≥ 3 CRS >50% PSA decline in 2 pts	[86]
NCT04104607	1	CC-1 (IgGsc)	PSMA xCD3	mCRPC	Recruiting		[87]
NCT04077021	1	CCW702 (DUPA)	PSMA xCD3	mCRPC	Recruiting		[88]
NCT05125016	1/2	REGN4336 + cemiplimab	PSMA xCD3	mCRPC	Recruiting		[89]
NCT04740034	1	AMG340 or TNB-585	PSMA xCD3	mCRPC	Recruiting		[90]
NCT05369000	1	LAVA-1207 (Gammabody^®^)	PSMA xVγ9Vδ2	mCRPC	Recruiting		
NCT03927573	1	GEM3PSCA	PSCA xCD3	PC, NSCLC, Renal cancer	Recruiting		
NCT04221542	1	AMG 509 (XmAb^®^) + Enzalutamide + Abiraterone	STEAP1 xCD3	mCRPC	Recruiting		[91]
NCT04702737	1b	Tarlatamab or AMG757 (HLE)	DLL3	NEPC	Recruiting		[92]
NCT03406858	2	HER2Bi-armed activated T cells + Pembrolizumab	HER2B xCD3	mCRPC	Recruiting		[93]

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
