# Peer review of "Bispecific T-Cell Engagers Therapies in Solid Tumors: Focusing on Prostate Cancer"

_cancers, 2023, doi:10.3390/cancers15051412_

Round 1
Reviewer 1 Report
The manuscript “Bispecific T-cell engagers therapies in solid tumors: focusing on
prostate cancer” by Simão et al, briefly introduces BitEs, their basic mechanism of action, and the associated adverse effects in the clinic (most adverse effects are common to other immunotherapeutic approaches such as CAR-T cells). The authors present current therapeutic approaches with BiTEs in solid tumors, to finally focus on their use in clinical trials for prostate cancer. Here, examples of BiTEs directed against prostate tumor-associated antigens are presented.
The description of the mechanism of BiTEs action is not thorough enough and the problems posed by the tumor microenvironment could also be addressed in more detail.
According to the title of the manuscript, section 4. “BiTEs in advanced prostate cancer”, should be the main one. The authors could consider to include there a subsection on "Neuroendocrine prostate cancer" and focus on DLL3 and KIT (KIT as a potential TAA for NEPC, see Cancer Cell 2022, 40: 1306–1323). Neuroendocrine prostate cancer is important enough to dedicate a brief section to it and introduce it properly since it usually derives from CRPC (Bluemn EG, et al. 2017 Androgen receptor pathway-independent prostate cancer is sustained through FGF signaling. Cancer Cell 2017, 32: 474).
Minor corrections:
· There are 3 references that the authors could consider including in this review due to their relationship with the topic,
o “Kfoury Y, et al., Human prostate cancer bone metastases have an actionable immunosuppressive microenvironment. Cancer Cell, 2021, 39: 1464–1478.”
o “Guan X, et al., Androgen receptor activity in T cells limits checkpoint blockade efficacy. Nature 2022, 606: 791–796.”
o “Su Y, et al., Targeting CD46 for both adenocarcinoma and neuroendocrine prostate cancer. JCI Insight 2018, 3: e121497.”
· Page 3. The following sentence mentions “costimulation” referring to TCR/MHC. Costimulatory receptors in T cells are CD28 and others alike, but not the TCR/CD3 complex. Please revise.
BiTEs are unique in their ability to redirect T-cells against TAAs on tumor cells and directly activate T-cells, independent of TCR/MHC costimulation [21].
· Page 3. The following sentence can be unclear to the reader.
Moreover, the activation of effector cells occurs only when both scFv binding domains are engaged with their respective target antigens [24].
The authors probably refer to “effector T cells”. Please specify which antigens are both scFv binding to.
· Page 4. The following expression “treatment infusion interruption” is unclear. Please revise.
In patients with mild CRS, supportive care is indicated, while grade ≥ 3 CRS is managed with treatment infusion interruption…
· Page 4. The following expression “adverse related “ is unclear. Please revise.
In addition, other relevant adverse related to the CD19 targeting mechanism have been reported with blinatumomab…
· Page 4. The following expression “extended elimination half-life “ is unclear. Please revise.
…incorporating an additional Fc region, creating a bsAb with a higher molecular weight and extended elimination half-life…
· Page 5. In the following paragraph the phenomenon of acquired resistance to BiTE therapy due to downregulation or loss of TAA, is addressed. The authors explain that:
…development of multiple novel T-cell engagers constructs, with different pharmacokinetic and pharmacodynamic profiles, including different formats with higher stability (dual affinity retargeting (DART®) bispecific antibodies), and some simultaneously targeting different TAAs…
The simultaneous targeting of different TAAs can solve the problem of acquired resistance. However, it is less clear how the use of dual affinity retargeting (DART®) bispecific antibodies, will solve the phenomenon of acquired resistance. Maybe the authors could explain it further.
· Page 5. Regarding section 3. BiTEs in solid tumors,
I suggest mentioning the reference below, related to the topic of the review: “Annu. Rev. Cancer Biol. 2022. 6:17–34, Targeting Solid Tumors with Bispecific T Cell Engager Immune Therapy.”
· Page 5. In the following sentence define TCB,
Specifically, in CEA-positive solid tumors such as metastatic colorectal cancer, RO6958688 (also known as CEA CD3 TCB or RG7802)
· Page 11. The following expression “metastatic-resistant prostate cancer (mCRPC).“ is unclear. Please revise.
… To date, sipuleucel-T is the only approved immunotherapy for metastatic prostate cancer yet is not considered a cornerstone therapy for men with metastatic-resistant prostate cancer (mCRPC).
· Page 14. The following expression “contains IgG antibodies“ is unclear. Please revise.
ES414/APVO4141/MOR209 is a bispecific antibody constructed with an ADAPTIR® format, which contains IgG antibodies
· Page 15. Define the following term “IgGsc“.
CC-1 is a bsAb targeting PSMA with a unique IgG-based structure. CC-1 was developed in an IgGsc format…
· References 28, 29. The web link of these references is not working. Please revise.
Some abbreviations need correction:
Page 2. Abbreviating "HD" is unnecessary, as it is not used anymore
High-dose interleukin-2 (HD IL-2), a potent inducer of cytotoxic T cells and NK cells…
Page 5. Abbreviating "(CiTEs)" is unnecessary, as it is not used anymore
To overcome this limitation constructs with concomitant immune-checkpoint action (checkpoint inhibitor T cell-engagers (CiTEs)) that target the PD-1/PD-L1 axis…
Author Response
We thank the reviewers for all the useful suggestions that have helped us to improve our manuscript. Please see the attachment.

Reviewer 2 Report
This review provides a nice perspective on BiTE therapies in solid tumours highlighting its limitations, obstacles, and innovations in prostate cancer.
I find the overall write-up solid. It is a well-written article and communicates the BiTEs technology to the broader scientific audience. I thoroughly enjoyed reading the review and the work captures relevant articles in the BiTE field. And, there is sincere effort involved in constructing the tables for BiTEs in solid tumours and prostate cancer with clinical trial information. I find that content very informative and highly relevant to the field and the community.
I have some minor concerns.
1. There are no figures/schematic representations for BiTEs presented in the paper. I would highly encourage the authors to include one or two figures that can make the entire reading process simpler for larger audiences.
2. Could you please describe in a few points how or why BiTEs (which are ScFv based) are better than small molecules or in particular heterobifunctional molecules?
3. Curious question, In BiTEs, there are linkers used to connect the ScFvs. How do they determine the linker type and linker size?
4. What should be the affinity of ScFv towards their POI and T Cell targets like CD-3? Does tweaking the affinity from strong to medium enable milder CRS or ICANS?
5. Though the immunotherapy treatment is limited to haematological malignancies, is there any particular reason, why the authors selected prostate cancer for this review? In other words, what are other solid tumours which may have better clinical benefits with BiTE like therapy?
I recommend this manuscript for publication with minor changes. Congratulations to the team.
Author Response

(The authors gave the same response as above.)

Reviewer 3 Report
General comment
The work by Simão and colleagues provides an extensive and interesting overview on bispecific T-cell engagers (BiTE) therapies in solid tumors.
After a general introduction and a section reporting the main results obtained in solid tumors, the review specifically addresses the results of BiTE therapies in prostate cancer. There’s an extensive listing of ongoing or complete clinical trials in this setting, and a section describing the possible and/or actual implementation of BiTE directed against several prostate cancer tumor antigens.
Since the main focus of the manuscript is prostate cancer, which is also mentioned in the title and is the element differentiating this work from other reviews on BiTEs already available in literature, a more detailed description of this specific clinical setting would help the reader appreciating the relevance of the clinical data reported. Therefore, we would suggest to include an extra section on prostate cancer, that should also include the challenges and hurdles (either unique or shared with other solid tumors) limiting the efficacy of BiTE therapy in this setting. Some of these information are already scattered in section 4 and 5 of the manuscript, but a more focused description would improve the quality of the Review.
Specific comments
In Table 1, please add wither the results or the “not reported” phrase for studies NCT03411915 and NCT04590781.
In section 5, please check the syntax of the sentence “The success of BiTE therapy in solid tumor malignancies will likely rely on two factors: 1) mitigation of severe toxicity and 2) proper selection of ideal identification (of what?).” Also make sure its sense does not sound contradictory vs the first sentence of the paragraph “The underlying causes of the limited effectiveness of BiTE therapy in solid tumor malignancies are multifactorial”: if there are multiple causes, are you sure its success will rely on 2 factors?
Author Response

(The authors gave the same response as above.)

Round 2
Reviewer 1 Report
The authors have substantially improved the manuscript.
In addition to an interesting description of current BiTE therapeutic approaches in solid tumors and prostate cancer in particular, the included tables are a useful update on clinical trials with BiTEs in prostate cancer.
Reviewer 3 Report
I thank the authors for revising the manuscript according to the reveiwers suggestions. I believe the quality improved and the manuscript can now be published as it stands